# Exploring convolutional KAN architectures with NAS

**Vladimir Latypov, Alexander Hvatov**
NSS Lab
ITMO University
Saint-Peresburg, 197101, Russia
{vlatypov,alex_hvatov}@itmo.ru

## Abstract

This study addresses the challenge of evaluating emerging neural architectures against extensively optimized legacy models. Kolmogorov-Arnold networks (KANs) offer a potential alternative to conventional deep learning, yet their benefits remain difficult to quantify. We introduce a neural architecture search (NAS) framework that systematically optimizes and compares KANs with convolutional neural networks (CNNs), eliminating human design biases. Experiments on image classification (MNIST, Fashion-MNIST, EuroSAT) and sea ice concentration estimation reveal distinct performance characteristics, demonstrating the impact of automated optimization on architectural selection.

## 1 Introduction

The pursuit of novel neural architectures has gained traction in recent years, particularly with the emergence of Kolmogorov-Arnold networks Liu et al. (2024b) (KANs). Although standard deep learning architectures, such as convolutional neural networks (CNNs) and multilayer perceptrons (MLPs), have undergone extensive optimization, it remains unclear whether new architectures like KANs can offer substantial advantages beyond incremental improvements. A significant challenge in such comparisons is eliminating biases introduced by expert-designed architectures and hyperparameter choices, ensuring a fair evaluation.

Several recent studies compare neural networks and KANs across different setups. Time series Genet & Inzirillo (2024), computer vision Cheon (2024a), PINNs and around Liu et al. (2024a), transformers Yang & Wang (2024). Most attempts and efforts are made to replace NN with KAN in a relatively standard setup (LSTM, PINN, transformer architectures). Less attention has been given to leveraging KAN's strengths in symbolic regression to better represent activation functions with different types of KAN architecture, like in Do et al. (2025).

There is a fair Shukla et al. (2024) and a fairer Yu et al. (2024) comparison that shows that KAN outperforms NN in symbolic regression. In contrast, NN outperforms all possible problems: table data, image processing with dense networks, NLP (natural language processing), and audio processing. However, many of these comparisons are based on manually selected architectures, making it difficult to assess whether the observed performance differences stem from the architectures or human-imposed design choices.

To address this issue, we introduce a NAS (neural architecture search) framework for a systematic and unbiased comparison of KAN and CNN architectures. By automating the search process, NAS minimizes the impact of human biases on model selection and hyperparameter tuning. We use this framework to evaluate both architectures on multiple standard benchmarks, including MNIST, Fashion MNIST, EuroSAT, and a real-world task: sea ice concentration prediction.

This study **aims** to demonstrate that NAS can reduce expert bias in architecture design, providing a more systematic evaluation of emerging neural architectures.

**Contribution:**

**Unified NAS Framework** – We introduce an evolutionary NAS algorithm that optimizes KAN and CNN architectures within the same search space, ensuring a fairer comparison.

**Empirical Benchmarking** – We evaluate NAS-optimized architectures on standard datasets and real-world tasks, providing a comprehensive assessment of their capabilities.

**Insights into KAN Performance** – We analyze the conditions under which KANs excel or struggle, particularly in convolutional settings, and discuss their limitations in deep architectures.

**Data and code** are available at anonymous repository `https://anonymous.4open.science/r/nas-kan-29A1`

**Limitations:** This study does not exhaustively evaluate all possible architectures and benchmarks. Additionally, the fairest comparisons require significant computational resources, which may limit scalability. We aim not to claim one architecture's absolute superiority over another but to highlight how expert bias in architectural design influences performance comparisons.

## 2  BACKGROUND

### 2.1  CONVOLUTION KAN LAYER GEOMETRIES

Kolmogorov-Arnold Networks (KANs) modify CNNs using learnable non-linear transformations instead of traditional linear filters, enhancing expressiveness and increasing computational complexity.

**KAN convolutions for images** In ConvKAN Bodner et al. (2024), the convolution kernel is defined as a matrix of learnable non-linear functions rather than fixed linear filters. Specifically, for an input value $x$, a KAN kernel transformation is given by

$$\phi(x) = w_1 \cdot \text{spline}(x) + w_2 \cdot \text{silu}(x) \tag{1}$$

, where $w_1, w_2$ are trainable weights that control the contribution of different non-linear transformations, $\text{spline}(x)$ is a learnable piecewise polynomial function that enables localized non-linear transformations, and $\text{SiLU}(x)$ is an additional non-linearity to enhance flexibility. CNN kernels are a special case where the spline function reduces to a linear mapping.

Like traditional CNN convolutions, a KAN convolution kernel scans the input feature map, applying non-linear transformations at each position. Unlike CNN kernels, which use fixed weights, KAN kernels apply learned functions, enhancing flexibility but increasing computational cost. Given an input image $a \in \mathbb{R}^{h \times w}$ and a KAN kernel $K$ of size $N \times M$ $K \in \Phi^{N \times M}$ where $\Phi$ is the space of spline functions, the convolution operation is defined as

$$(\text{Image} * K)_{i,j} = \sum_{k=1}^{N} \sum_{l=1}^{M} \phi_{kl}(a_{i+k,j+l}) \tag{2}$$

,where $\phi_{kl}$ is the learnable mapping applied at position $(k, l)$ within the kernel.

**Extension to multiple channels** An important architectural choice in KAN-based convolutional networks is the connectivity pattern between input and output channels. We refer to it as a map from input $c_{\text{in}}$ and output $c_{\text{out}}$ channels where a $c$ channel feature map is of the form:

$$\text{FeatureMap} \in \mathbb{R}^{c \times N \times M}$$

We consider two main approaches for extension:

**Sparse KAN convolutions** (ConvKAN-style) compute each output channel from a single input channel, leading to a grouped convolution structure where different channels process separate aspects of the data. The convolution kernel for a sparse KAN layer is

$$K \in \Phi^{n_{\text{convs}} \times N \times M}$$

Each of the elements of $c_{\text{out}} = c_{\text{in}} \cdot n_{\text{convs}}$ output channel feature map depends on the corresponding pair of kernel and input channel taken from their dot product:

$$(\text{FeatureMap} * K)_{sn_{\text{convs}}+c,i,j} =$$

$$\sum_{k=1}^{N}\sum_{l=1}^{M} \phi_{ckl}(a_{s,i+k,j+l}) \quad (3)$$

, so each output channel only depends on one input channel.

Sparse KAN convolutions, like standard grouped convolutions, restrict feature interactions, which may limit representation power in deeper networks. **Dense KAN convolutions** mitigate this limitation by allowing full connectivity across channels, improving hierarchical feature extraction. The kernel formulation is

$$K \in \Phi^{c_{\text{out}} \times c_{\text{in}} \times N \times M}$$

, and the formula for FeatureMap convolution is:

$$(\text{FeatureMap} * K)_{c,i,j} = \sum_{c'=1}^{c_{\text{in}}}\sum_{k=1}^{N}\sum_{l=1}^{M} \phi_{c'ckl}(a_{c',i+k,j+l}) \quad (4)$$

, which means that each output channel depends on all input channels.

Throughout this paper, we refer to the layer with Eq. 4 connection pattern as dense KAN convolutions. We will use the general term KAN convolution to refer to both of these input-output channel connectivity patterns.

In standard convolutional terminology, sparse KAN convolutions correspond to group convolutions groups $= c_{\text{in}}$, whereas dense KAN convolutions correspond to depth-wise convolutions with complete mixing groups $= 1$.

**Transposed KAN convolutions** In applications requiring upsampling, such as image generation and time-series forecasting, transposed convolutions are needed. To extend KAN convolutions to transposed variants, we define the operation as

$$(\text{FeatureMap} * K)_{c,i,j} = \sum_{\substack{c',r,s,k,l \\ r+k=i \\ s+l=j}} \phi_{c'ckl}(a_{c',i+k,j+l}) \quad (5)$$

, ensuring that non-linear transformations are applied symmetrically during upsampling.

**Overall model structure** Every model considered in the paper is a series of interconnected convolutional layers of CNN or KAN convolutions with different geometries and KAN or NN fully connected layers. Every layer has a parameter set, such as input size, kernel size, stride size, and weights.

It is convenient for the NAS application to consider two different parts of the model - structure $S = \{s_1, s_2, ...s_j\}$, where $s_i$ , $i = 1, .., j$ is the layer type and data flow encoding and parameters set $P = \{p_1, ...p_j\}$, where $p_i$ , $i = 1, ..., j$ is the corresponding layer's numeric or categorical parameter set. So, the resulting model is considered in the form

$$M(S, P, x) : X \rightarrow Y \quad (6)$$

In Eq. 6, $x \in X$ is an element of input space $X$, and $Y$ corresponds to the problem target set.

## 2.2 NAS AT A GLANCE

Neural Architecture Search (NAS) optimizes performance for specific tasks. NAS methods can be roughly categorized into three approaches: NAS from scratch, full NAS, one-shot NAS, and few-shot NAS Poyser & Breckon (2024), each with distinct methodologies, strengths, and limitations. Roughly, we depict the scheme in Fig. 1. We do not intend to make a full review of each method of NAS, but we try to describe the requirements to start with a novel architecture.

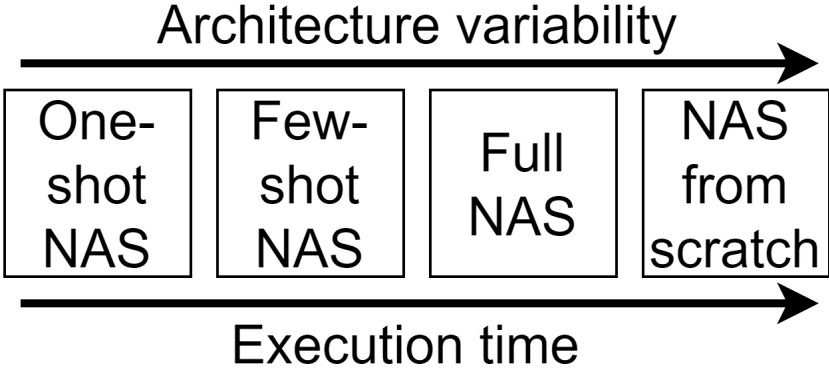

Figure 1: NAS classification scheme

The difference between NAS algorithms is the building blocks for architectures: basic operation, basic layers, and parametrized or fixed layer combinations. The smaller the scale of the block, the more variability of architecture is possible, but it is achieved with a larger number of intermediate steps.

**NAS from scratch** focuses on generating neural architectures and their weights entirely from the ground up Geada et al. (2024); Ericsson et al. (2024), without relying on pre-trained models, weight inheritance, or structured priors. This approach simultaneously explores the search space and the training process, optimizing the architecture and its parameters from an uninitialized state.

**Full NAS** (for example, Tong & Du (2022)) conducts an exhaustive search of the architectural space to identify the optimal network configuration. Traditionally, this method employs reinforcement learning, evolutionary algorithms, or Bayesian optimization to guide the search process. Full NAS explores the search space thoroughly, often yielding architectures with state-of-the-art performance. Multiple objectives can also be used, such as accuracy, latency, and energy consumption. However, the search process can take weeks or even months on large-scale datasets.

**One-Shot NAS** (also often referred to as weight sharing, for example, ENAS Pham et al. (2018)) addresses the computational challenges of full NAS by training a single, over-parameterized "supernet" that encompasses all possible architectures. Candidate architectures are then evaluated as subnetworks of the supernet without independent retraining. However, supernet architecture must be defined so that it can be reduced thereafter. Sub-networks often inherit weights from the supernet, leading to discrepancies between search time and stand-alone performance.

**Few-Shot NAS** (as an example Liu et al. (2018a)) serves as a middle ground between the full NAS and the one-shot NAS, using partial training of candidate architectures or transfer learning to reduce computational demands using an intermediate assessment model. It often involves training a smaller pool of architectures for a limited number of epochs or fine-tuning pre-trained networks. The drawback is that the smaller pool of candidates may lead to sub-optimal architectures compared to full NAS.

Our **aim** is to show that the standard CNN approaches to build architectures may not apply to convolutional KANs in the general case. However, we already have some prior blocks, such as splines, and a way to form convolution using splines. To compare CNN and KAN with convolution layers

(it could not necessarily be a KAN, any novel architecture) in a fair way, we are required to generate architecture from small but existing priors and compare the best possible result within a uniform setup. Therefore, we require (a) a full NAS algorithm to have as broad a search space as possible and (b) a NAS algorithm with the ability to work simultaneously with CNN and convolutional KAN. Below, we briefly describe the evolutionary Full NAS approach used in the paper.

## 3   Used NAS Approach

Among different search algorithms – gradient-based, RL, and evolutionary – for novel architecture, the most viable competitors are the last two. Gradient-based methods (DARTS-like Liu et al. (2018b)) require a fixed-dimensional search space, where inactive parameters are often set to zero. The fixed numerical search space may introduce an implicit bias in the search process. We use a relatively standard evolutionary multi-objective optimization approach based on the MOEA/DD algorithm with the following features:

– Models are represented by DAGs (directed acyclic graphs) with single sink
– Computation volume is constrained
– Evolutionary operations are made such that the DAG structure is preserved

In this case, DAG is represented by parametrized convolutional and fully connected layers, as vertices and edges represent data flow. Regarding Eq. 6, DAG is the structure $S$. The parameters set is determined during the training of a model with a given structure.

As a result, optimization has the form:

$$\arg\min_{S\in\Sigma} F(M(S, P, x)) \tag{7}$$

In Eq. 7 $F(\cdot) = F(f_1, ... f_q)$ is the mutli-objective operator over objectives $f_i$, $\Sigma$ is the set of all achevable structures. For optimization we use objective $f_{\text{metric}} = \min_P \mathcal{L}(M(S, P, X), Y)$, where $\mathcal{L}$ is the loss function corresponding to a given problem and objective $f_{\text{par}}(S) = |P|$ with $|P|$ is the number of parameters for a given structure $S$.

More details and a complete realization scheme can be found in Appendix A.

## 4   Experimental Setup

### 4.1   Approach to comparison

The guiding principle of the experiments is to ensure that the evolutionary process is as exhaustive as possible. However, it is impossible to cover every point in the search space due to the full computational complexity of the NAS. The following measures have been adopted to achieve as full a search space as possible:

- The search space is narrowed by introducing inductive biases specific to each type of structure (CNN, KAN, CNN+KAN).

- Smaller models are discovered to save computational resources, allowing for a larger number of trials, and the corresponding more manageable benchmark datasets are selected to suit small models' evaluation better.

- The models are compared based on benchmark metrics and complexity.

- Complexity is computed as the number of learnable parameters of the model.

While limiting model size reduces computational cost, it may also restrict the discovery of high-performing architectures. We mitigate this by allowing broader kernel size variations and flexible layer depths.

The full justifications for the particular choices are available in Appendix B.

## 4.2 DATASETS

For classification, standard benchmarks were taken:

**MNIST** (LeCun et al. (2010) available under CC-BY-SA 3.0 license)
**Fashion MNIST** (Xiao et al. (2017) available under MIT license)
**EuroSAT** (Helber et al. (2019) available under MIT license)

Another non-benchmarked problem was taken to avoid bias: sea-ice concentration prediction based on the **OSI SAF** Global Sea Ice Concentration (SSMIS) data Lavergne et al. (2019) (available under the CC-BY 4.0 license). The data were adjusted to a consistent 14 km grid, hence being representable as an image. The Laptev Sea is selected for the experiments. The data's temporal resolution was reduced to a week to facilitate long-term forecasting and minimize insignificant dynamic reproduction. Forecasting for one year is performed using data from the previous two years. Such a 3-year window is sliding through the timeline to generate the dataset.

## 4.3 EXPERIMENTAL PARAMETERS

The common parameters chosen for experiments are listed in Tab 1.

Table 1: Common hyperparameters used in NAS

| Parameter | Value |
|---|---|
| n_generations | 7 |
| pop_size | 8 |
| Pooling kernel_size & stride | 2 |
| max_parameters | $< 200k$ (low range), $< 1.5m$ (high range) |
| NN train_epochs | task dependent |
| NN train timeout (min) | task dependent |
| Convolution kernel_size | {3, 5 } (KAN) {3,5,7}(CNN) |

While the multi-objective approach to evolution allows sharing of fitness information between networks of different size categories, a more reliable approach is to merge multiple evolution algorithm runs for different parameter count ranges. In Tab 1, merged ranges are referred to as `max_parameters`.

For NSGANet Lu et al. (2019) for `search_space=micro` the suggested parameters are `n_generations=30` and `n_individs=20`. In our case, the search space was wider, and KANs, unlike their NN counterparts, have extended train time. We determine parameters such as the plateau reached for the Pareto frontier and training loss to preserve fairness.

The maximum number of epochs is set to 30, with a timeout of 12 minutes for the MNIST and FashionMNIST datasets, and 50 epochs with a timeout of 40 minutes for the EuroSAT dataset.

A larger number of epochs is employed for the final solution refinement: 50 epochs for the MNIST and FashionMNIST datasets and 70 epochs for the EuroSAT dataset.

### 4.3.1 CNN SETUP

BatchNorm2d, pooling, and activations of the following types are employed in the convolutional layers as node supplementary operations (Appendix A.4): ELu, SeLu, SoftPlus, ReLu, SoftSign, tanh, HardSigmoid, and Sigmoid.

The number of convolutional layers, n_conv_layers, ranges from 2 to 8 for the EuroSAT dataset and from 2 to 6 for the MNIST and FashionMNIST datasets for both parameter ranges. The convolutional filter depths, conv_filter_depths, are the same for all datasets and range from 16 to 64 for the high parameter range and 4 to 64 for the low parameter range.

The number of fully connected layers, n_fc_layers, ranges from 2 to 3 for all datasets. Their hidden sizes range from 32 to 128 for the high parameter range and 16 to 128 for the low parameter range.

### 4.3.2 KAN Setup

The extremely small and large values of parameters have been demonstrated in ConvKAN to be less effective and do not align with the principle of exhaustiveness (Appendix B.1). Specifically, spline_order = 3 and grid_size is always one of $\{5, 10\}$. Consequently, the search space is restricted to these choices for both fully connected and convolutional KAN layers.

For grayscale datasets, the convolutional filter depths, conv_filter_depths, range from 2 to 16, while for the EuroSAT dataset, they range from 3 to 24. These choices are consistent across KAN convolutional layer geometries and parameter ranges.

Following the authors of ConvKAN, we only consider one fully connected KAN layer. This decision is based on the fact that having multiple layers in the classification head of a KAN causes an explosion in the number of parameters and computational resources, placing it outside of the concerned search space. The reasons for the high number of parameters in this case are as follows:

- Even with the same dimensions, the KAN Linear layer contains more parameters than a CNN layer.
- Multiple layers in the classification head imply that the preceding layers are larger than the number of classes, making the number of parameters for a multilayered head even larger and the first layer the largest parameter consumer.
- A typically lower number of KAN convolutions than CNN convolutions results in a large matrix size in the final convolution layer and, hence, a large output from the flattening layer.

However, the complexity of the classification head can be controlled by the choice of grid_size.

### 4.4 Time series forecasting

Lagged transformation (sliding window) is applied to the test and train datasets to collect sufficient dataset sizes. Transformations are not applied since only rotations by $90° \cdot k$ are physical and impractical since the cross-sea transfer learning is not conducted.

Using layers with `padding=same` could lead to non-physical prediction. Sea ice has spatial distribution and is governed by physical laws that should be preserved – mechanics, heat transfer, and others. Hence, transposed convolutions are used.

The total number of convolutions and transposed convolutions ranges from 4 to 8 in the CNN case and 2 to 4 in the KAN case. The number of steps in the logarithmic scale between output_size and input_size for channel numbers of convolutional layers is 10.

## 5 Experimental Results

As contestants we consider the following search spaces for NAS: `CNN`, pure convolutional NN, `KAN`, following ConvKAN, such models use KAN convolutional layers with ConvKAN geometry and fully connected KAN layers in classification head, `CNN+KAN`, the same KAN classification head follows classical convolutions, `denseConvKAN`, dense KAN convolutional layers (Eq. 4) with KAN classification head.

Where applicable, we provide results reported in ConvKAN Bodner et al. (2024) paper with mark (ConvKAN).

For all experiments single `NVIDIA A100` GPU with 48 GB RAM within `DGX A100` server was used.

For every type of network, the Pareto frontier was recorded together with optimization history and a complete model set in population. Every model of the resultant Pareto fronts has been evaluated ten times to account for the stochasticity of the learning process.

**MNIST** is usually considered a straightforward problem. However, it shows that the approach generally works. In Fig. 2, the final Pareto frontier, together with the confidence interval and distributions, is shown.

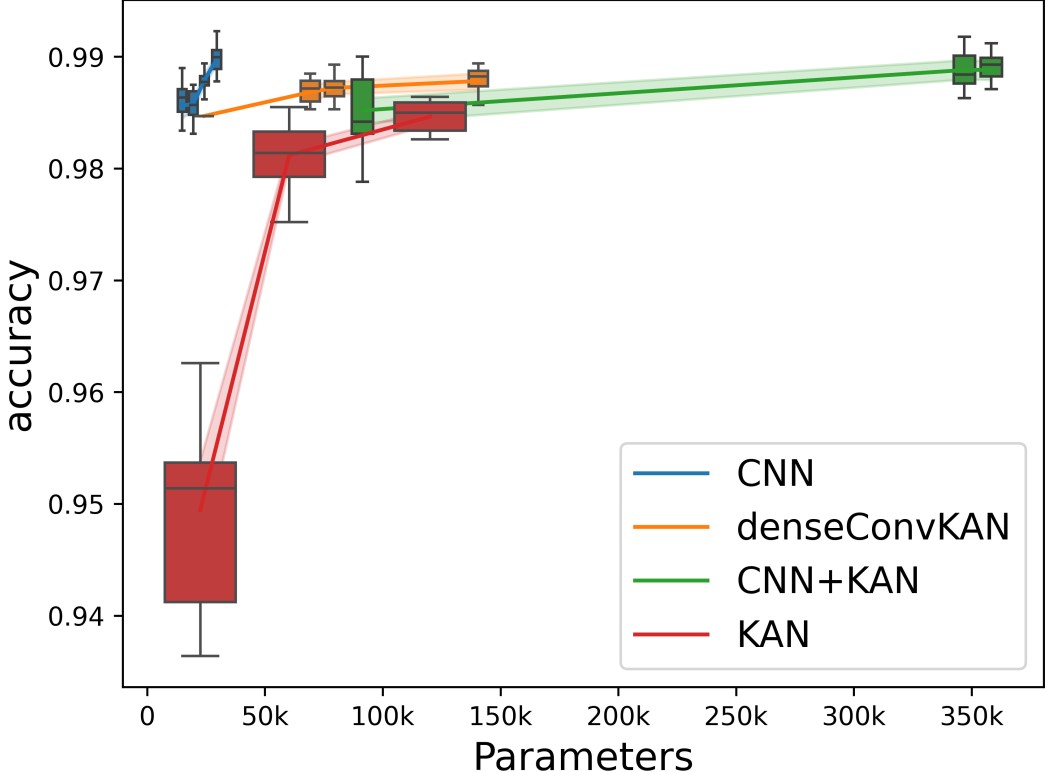

Figure 2: MNIST Pareto frontier. Boxes are accuracy distributions for a given parameter range. Box width is proportional to the considered parameters range. Lines are mean Pareto frontiers with 95-% confidence interval across ten runs.

The best model accuracy for the MNIST dataset and the number of parameters is shown in Tab. 2.

Table 2: MNIST best models discovered

| Architecture | #Parameters | Accuracy |
|---|---|---|
| CNN | 29.6k | **0.992** |
| CNN (ConvKAN) | 157.0k | 0.991 |
| denseConvKAN | 140.6k | 0.989 |
| CNN+KAN | 346.9k | **0.992** |
| CNN+KAN | 91.6k | 0.99 |
| CNN+KAN (ConvKAN) | 95.0k | 0.988 |
| KAN | 120.1k | 0.986 |
| KAN (ConvKAN) | 94.9k | 0.989 |

The table shows that a more competitive accuracy (0.991 vs. 0.989) may be obtained with fewer parameters (157.0k vs. 140.6k). However, CNNs could not be beaten with KAN architectures for the best possible accuracy. We also mention that the human-built model reported in Drokin (2024) achieves a validation accuracy of 99.48 with the number of parameters 1542199 (1.5m).

**Fashion MNIST** is considered an alternative to MNIST but with more complex content. Pareto frontiers obtained during NAS are shown in Fig. 3.

The best model accuracy for the Fashion MNIST dataset and the number of parameters is shown in Tab. 3.

The paper Bodner et al. (2024) reports the best accuracy of $0.8856$ for `KKAN` (which is considered the direct human-built competitor to denseConvKAN) architecture with $74.88k$ parameters. We can achieve an accuracy of $0.905$ with a double number of parameters $164.6k$.

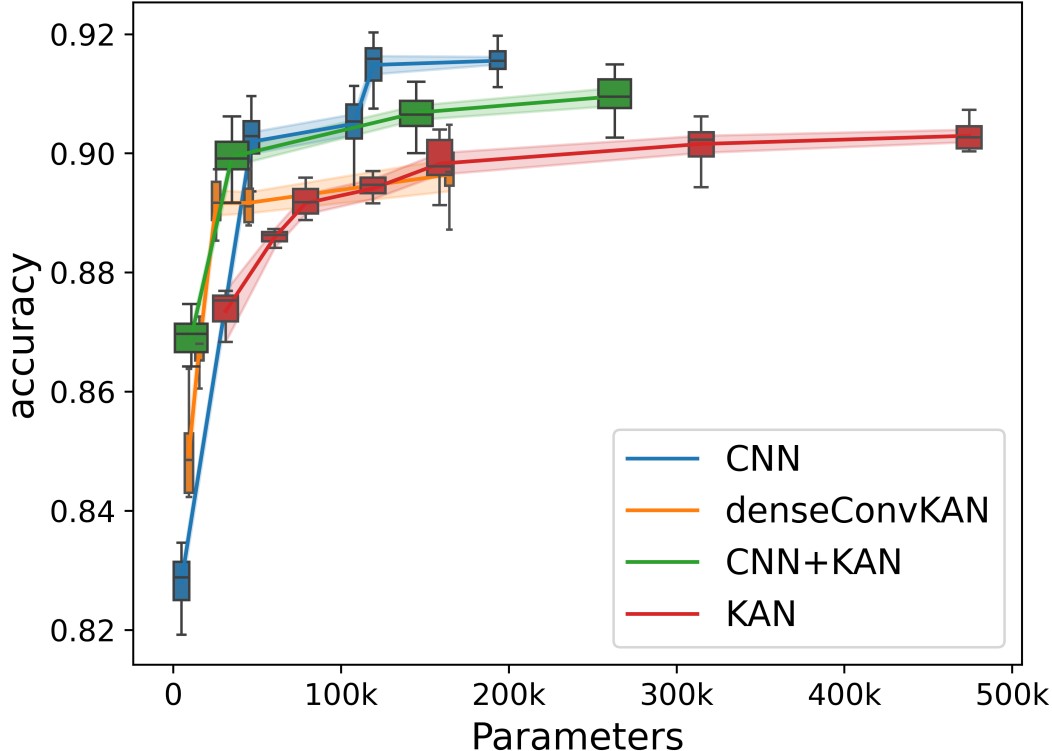

Figure 3: FashionMNIST Pareto frontier. Boxes are accuracy distributions for a given parameter range. Box width is proportional to the considered parameters range. Lines are mean Pareto frontiers with 95-% confidence interval across ten runs.

Table 3: FashionMNIST best models discovered

| Architecture | #Parameters | Accuracy |
|---|---|---|
| CNN | 119.4k | **0.92** |
| CNN | 46.6k | 0.91 |
| CNN (ConvKAN) | 157.0k | 0.901 |
| denseConvKAN | 164.6k | 0.905 |
| CNN+KAN | 263.2k | 0.915 |
| CNN+KAN | 35.1k | 0.906 |
| CNN+KAN (ConvKAN) | 95.0k | 0.888 |
| KAN | 474.4k | 0.907 |
| KAN | 78.9k | 0.898 |
| KAN (ConvKAN) | 94.9k | 0.897 |

**EuroSAT** We could not find a human-built convolutional KAN architecture. However, it is often used in NAS algorithms. The resulting Pareto frontier is shown in Fig. 4.

The best models for EuroSAT are shown in Tab 4.

Table 4: EuroSAT best models discovered

| Architecture | #Parameters | Accuracy |
|---|---|---|
| CNN | 214.9k | **0.932** |
| denseConvKAN | 210.1k | 0.918 |
| CNN+KAN | 296.3k | 0.922 |
| KAN | 922.3k | 0.805 |

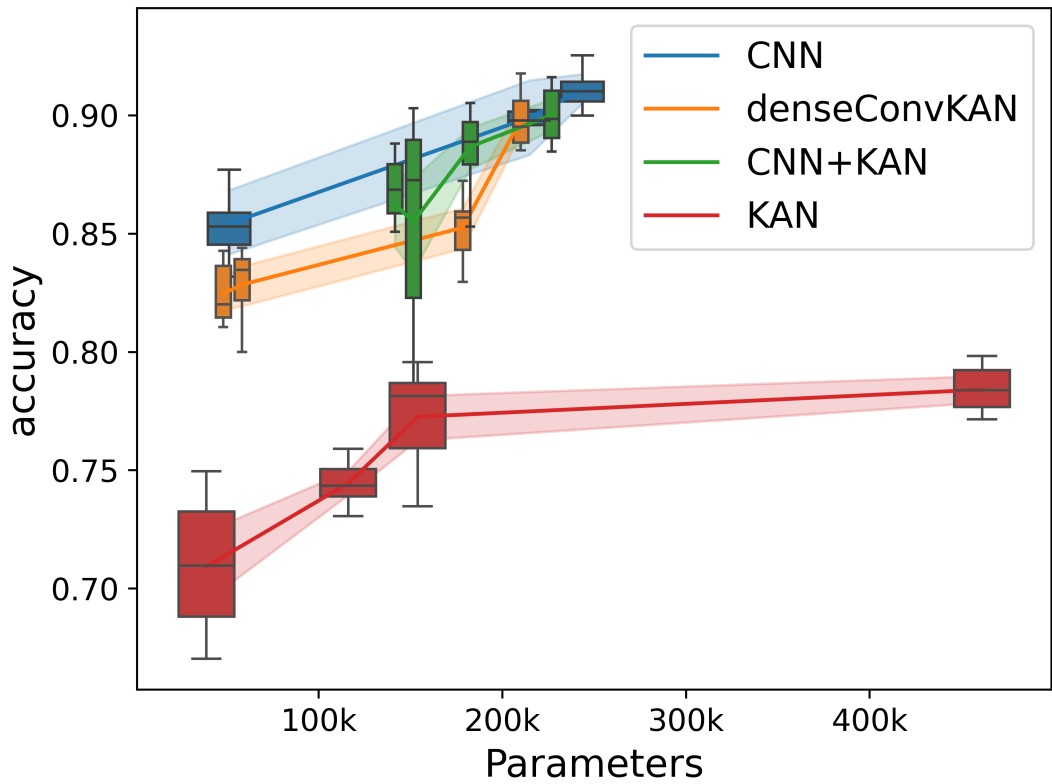

Figure 4: Eurosat Pareto frontier. Boxes are accuracy distributions for a given parameter range. Box width is proportional to the considered parameters range. Lines are mean Pareto frontiers with 95-% confidence interval across ten runs.

Although there are no direct competitors, we refer to Cheon (2024b) reporting that ConvNeXt with MLP replaced to KAN (CNN + KAN direct competitor) has 94 % precision with an amount of parameter ranging from 29m to 200m for ConvNeXt Liu et al. (2022) with MLP. In our case, `denseConvKAN` has the same performance as CNN+KAN with fewer parameters.

**OSI SAF Sea-ice (SSMIS)** As a task unbiased by standard benchmarking, we use sea ice concentration data at the Laptev Sea. The main difference is that we require transposed convolutions introduced in the pipeline to make forecasts. Two samples of prediction are shown in Fig. 5.

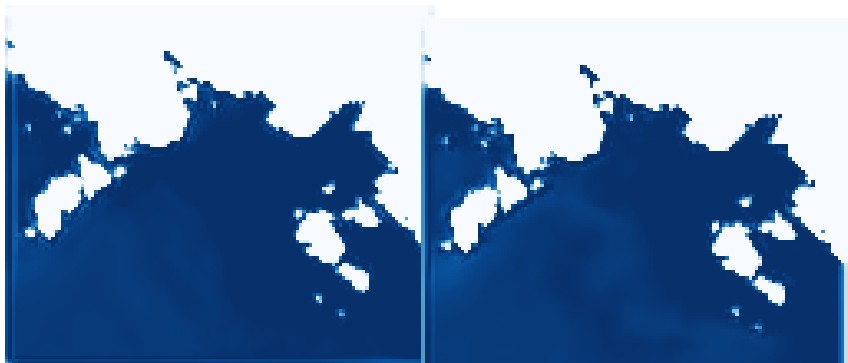

Figure 5: Laptev sea ice prediction. The first week (left) and last (52, right) of prediction for one year are shown.

The resulting Pareto frontier is shown in Fig. 6.

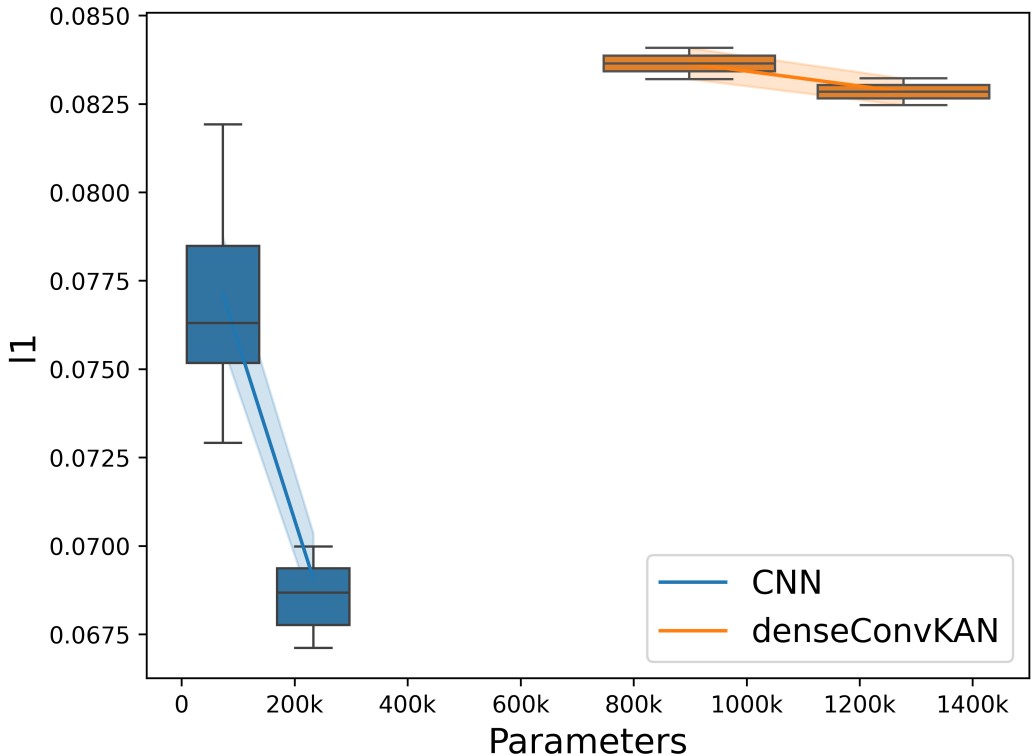

Figure 6: Laptev sea Pareto frontier. Boxes are accuracy distributions for a given parameter range. Box width is proportional to the considered parameters range. Lines are mean Pareto frontiers with 95-% confidence interval across ten runs.

The best prediction models are shown in Tab 5.

Table 5: Ice prediction in the Laptev Sea best models discovered

| Architecture | #Parameters | L1 Loss | SSIM |
|---|---|---|---|
| CNN | 233.2k | **0.067** | **0.79** |
| CNN | 97.0k | 0.069 | 0.778 |
| CNN | 73.3k | 0.073 | 0.773 |
| CNN | 73.2k | 0.074 | 0.768 |
| denseConvKAN | 1.3M | 0.082 | 0.74 |
| denseConvKAN | 898.6k | 0.083 | 0.74 |

KAN models may struggle in this task due to the lack of strong spatial priors, which CNNs inherently exploit through weight sharing in convolutional layers. However, the CNN results might have sharp edges and not satisfy physical constraints. So, the resultant fields of the models are hand-reviewed after the evolutionary process.

## 6 DISCUSSION

**Deep architecture design with ConvKAN layers**    Constructing deep architectures with classical ConvKAN geometry is unfeasible because adding more convolutions results in a power-law growth of elements , causing the convolutional weights' share to decrease rapidly with depth. This reduces spatial data bias from convolutional layers, worsens quality, and causes computational inefficiencies because ConvKAN's geometry does not mix channels until the fully connected part, leading to replication.

Even in smaller models, frequent pooling layers are needed to manage feature map size by reducing spatial resolution, unlike in classical CNNs, where pooling follows a block of convolutions. Calculations in Table 6 show ConvKAN models like KANConv&MLP and KKAN have low parameter fractions in convolutional layers.

Table 6: Convolution and linear layer percentage

|  | #Parameters | FC, % | Conv, % |
|---|---|---|---|
| 1 Layer MLP | 7.8k | 100.0% | 0.0% |
| ConvNet (Small) | 2.7k | 89.78% | 10.22% |
| ConvNet (Medium) | 157.0k | 99.24% | 0.76% |
| ConvNet (Big) | 887.5k | 90.77% | 9.23% |
| KANConv&MLP | 163.7k | 99.45% | 0.55% |
| Simple Conv&KAN | 37.0k | 99.24% | 0.76% |
| KKAN | 94.7k | 99.05% | 0.95% |
| ResNet152 | 58.2M | 5.06% | 94.94% |
| ConvNeXt (small) | 49.5M | 0.02% | 99.98% |
| AlexNet | 57.0M | 95.67% | 4.33% |
| MobileNet | 2.2M | 0.57% | 99.43% |

Despite the extremely low share of convolutional part parameters, the amount of computations of the convolutional part is comparable to regular CNNs due to a higher params-to-flops ratio.

**Other Approaches to Evolution**  It should be noted that more refined approaches could be employed instead of the NAS algorithm used to accumulate more abstract information on the dependence of the quality-complexity trade-off resolution on the position in the graph. This leans towards indirect encoding or surrogate models, i.e., the Few-Shot NAS, which is beyond the scope of this paper.

A carefully designed algorithm of this kind can function as an end-to-end algorithm. It could efficiently identify the subspaces to search for each type of structure when running on a larger shared space, leading to even less biased results by further reducing human intervention compared to regular NAS and allowing for analyzing big models without loss of exhaustiveness.

**Dropout Layers**  In terms of KANs, there are no changes in the dropout requirement to avoid overfitting, compared to CNNs/MLPs. The DropKAN study Altarabichi (2024) proposes a version of the Dropout layer adjusted for the KAN architecture.

However, in CNNs, batch normalization has largely supplanted dropout, performing its functions and being somewhat incompatible with it Li et al. (2019). Consequently, this type of layer is not included in the search space.

**Concatenation Across DAG Input Edges**  A common approach in DAG-based NAS is concatenating all incoming edges to obtain the node input, as seen in manually designed architectures such as inception blocks. However, the encoding used in this study only supports merging activations by an element-wise sum like that in ResNet. This encoding is less rich but enables one to maintain exhaustiveness.

## 7   CONCLUSION

NAS allows for better model performance for KAN, yet it remains unfeasible for real problems due to the extended training time. With the proposed approach, we were able to create more appropriate building blocks for convolutional networks within the KAN paradigm.

The main findings are the following:

- The fairest conclusion is that KANs achieve **comparable** accuracy to CNNs on image classification tasks but require significantly more computational resources.

- To build KAN architecture with convolutions, modifications such as dense KAN convolutions and transposed convolutions are required.

- By automating architecture discovery, NAS reduces human bias in model selection, revealing alternative designs that might otherwise be overlooked.

Our results suggest that KANs can achieve comparable accuracy to CNNs but require significantly more computational resources. This study highlights the importance of NAS in fairly evaluating novel architectures, ensuring that promising designs are not overlooked due to human biases in architecture selection.

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

## A    EVOLUTIONARY NAS

Due to the high cost associated with the objective evaluation in Neural Architecture Search (NAS), the $(\lambda + \mu)$-EA evolutionary algorithm is utilized to enhance exploitation.

### A.1    EVOLUTIONARY ENCODING OF NETWORKS

A special type of Directed Acyclic Graphs (DAG) with a single source, a single sink, and node input arity (i.e., the number of incoming data sources to a computation node) of 1 or 2, where the second input edge serves as a shortcut connection, is utilized as an encoding, as illustrated in Figure 7 and Figure 8. The graph nodes are treated as layers, and the edges are data paths between layers. So, if the input of a node is 2, the input of the corresponding mapping is calculated as an element-wise sum of the two connected node outputs. The input image is passed as the input to the source node, and the result is taken from the sink node. A model is rendered from the graph, learned, and evaluated on the validation set to evaluate an individual in the fitness function.

For both classification and time-series forecasting tasks, the transformations of the convolutional part are applied first.

For classification (Fig, 7), a single flatten layer is the part of the model following the convolutional backbone and preceding the fully connected part (classification head). To guarantee that the output shape of the last layer of the model matches the number of classes, an additional last layer of the corresponding shape is added to the model during its construction from the graph. Cross-entropy holdout validation is implemented as the fitness function.

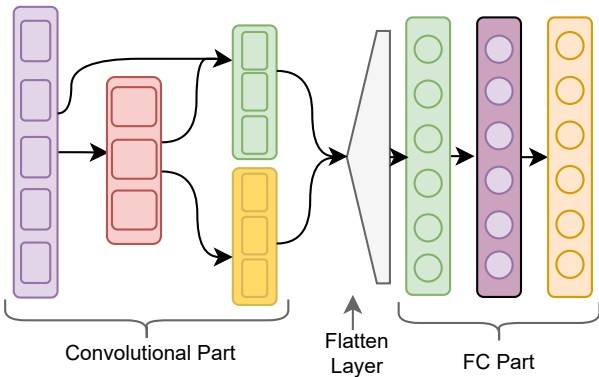

Figure 7: NAS encoding for classification tasks

For time series forecasting (Fig. 8), the convolutional backbone consists of two parts with the same number of layers along any data path: encoder and decoder (see Section 4.4). The output feature map of the sink node is interpreted as the resultant sequence of images evaluated against the ground truth of the validation set with the $l1$ metric.

Hyperparameters for each node, such as the number of filters, kernel sizes, activation types, spline grid sizes, etc., are chosen individually rather than uniformly across the entire graph. Each node negotiates a trade-off between quality and simplicity, the resolution of which depends heavily on its position within the graph.

Fully connected layers (KAN and MLP), CNN convolutional layers, and dense KAN convolutional layers in the classification task are sized as powers of 2 within a specific range, producing a reasonable logarithmic scale. A logarithmic scale with a smaller factor is used for time series forecasting to cover the sizes between the number of input channels (max) and output channels (min).

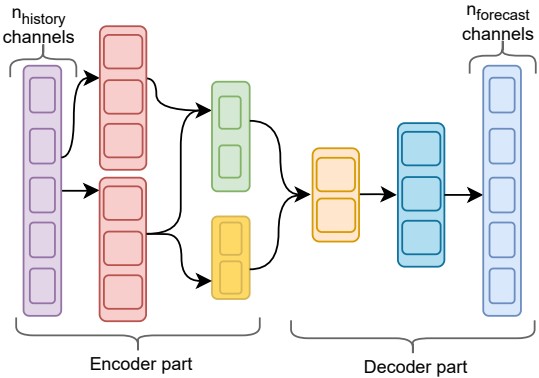

Figure 8: Time series forecaster NAS encoding

According to the specialized data flow in convolutions of ConvKAN's geometry, the output dimension of this type of layer should be divisible by its input dimension. Therefore, dimensions for such layers are selected from the form image_input_channels $\cdot\, 2^k$, where $k \in [l, r]$.

## A.2 GENETIC OPERATORS AND VALIDATION RULES

### A.2.1 GENETIC OPERATORS

During the application of each genetic operator (mutation and crossover), changes are iteratively sampled using mutation and crossover attempts. The first change that satisfies the *validation rules* is accepted. Mutation types include adding a single node, dropping a single node, adding a single edge, and changing a node, while crossover involves exchanging subtrees within the graphs. For time series forecasting, due to the rigid encoder-decoder structure of the model (Section 4.4), a pair of consecutive mutations is sampled before the validation rule check because any single mutation changing the model structure would affect only one of the two parts and thus would violate the equality constraint on the number of layers of encoder and decoder along each data path. This makes the search space connected.

### A.2.2 VALIDATION RULES

Validation rules stipulate that the graph must remain a DAG with a single source and sink and that proper separation between network parts (convolution, flatten, classification head for classification and encoder, decoder for time series forecasting) is maintained, along with dimension correctness. Less obvious rules include:

- Convolution filter depths must increase along any data path in classification tasks and decrease for time series forecasting from input channels (= given period length) to output channels (= forecasting period length).

- Complexity-controlling rules to allow "thin" architectures (see A.6).

### A.2.3 SEARCH SPACE CONNECTEDNESS

Given the constraints imposed by the validation rules, one might question whether the search space remains connected because any model within the search space can be reached from any other model through a sequence of these mutations.

### A.3 BATCHNORM

The primary function of the Batch Normalization (BatchNorm) layer is to adjust data into the non-trivial range of activation functions. However, the KAN layers' grid extension capability already achieves this effect.

In ConvKAN, batch normalization was experimentally introduced as an alternative to the built-in KAN scaling, but no significant improvement was observed.

Consequently, in our experiments, the BatchNorm layer is applied only to CNN convolutions and not to KAN convolutions for both theoretical and practical reasons.

### A.4 SUPPLEMENTARY OPERATIONS IN A NODE

In addition to the main convolutional operations, each convolutional node of the graph incorporates several *supplementary* operations that are sequentially applied during the forward pass. When generating new nodes through mutations, supplementary operations are added with a certain probability.

The supplementary operations include batch normalization, activation, and pooling, where applicable.

The set of supplementary operations considered and their associated probabilities depend on the type of layer, as described in Section 4.3.

An alternative approach would be to treat these operations as separate graph nodes. However, this would complicate the design of mutations necessary to maintain the layered structure with alternate convolutions and supplementary operations, which structure is known to be effective. Mutations might otherwise disrupt this structure.

### A.5 MULTIOBJECTIVE

In ConvKAN, models are manually compared based on the number of parameters and quality metrics across a set of manually constructed models. When applying Neural Architecture Search (NAS), a natural extension explicitly searches for this trade-off as a Pareto frontier, as typically done in multi-objective optimization MOEA/DD, with the number of parameters serving as the second metric.

Multi-objective optimization serves a dual purpose. First, the Pareto frontier is helpful for presenting results. Second, it helps overcome local minima by allowing intermediate models to survive due to their smaller size. Consequently, the multi-objective approach can achieve better fitness function values than a single-objective algorithm.

The particular complexity metric choice is discussed in B.2.

### A.6 COMPUTATIONAL VOLUME CONSTRAINT

A strict computational volume constraint is imposed during the optimization process to diversify the search space and narrow it to models with a low computational volume.

Simultaneously, constraints on specific model parameters that affect complexity, such as the maximum number of layers and filters per layer, should be higher.

The rationale is to widen the search space to permit deep and "thin" models; however, if every parameter is maximized, the computations will become excessively lengthy.

This approach is expected to yield more interesting models while conserving computational time.

Therefore, a small percentile of the generated models, which would otherwise consume a substantial portion of computational resources, is truncated.

Computational volume is measured by repeatedly passing a random tensor of batch dimensions through a model and recording the time.

The computational resource thresholds are experimentally determined to exclude only a small percentage of the generated individuals.

Similarly, a parameter number constraint is imposed to maintain exploration within the search space delineated in Section B.4. The threshold for parameters is derived from the considerations discussed in that section.

## A.7 STATISTICAL EVALUATION

Experimentation reveals that the stochasticity of learning a model on the training dataset may affect the rankings. This is acceptable for the evolutionary process due to its stochastic nature and constraints on fitness evaluation time. However, more accurate results may be obtained for the final best individuals.

We use 95

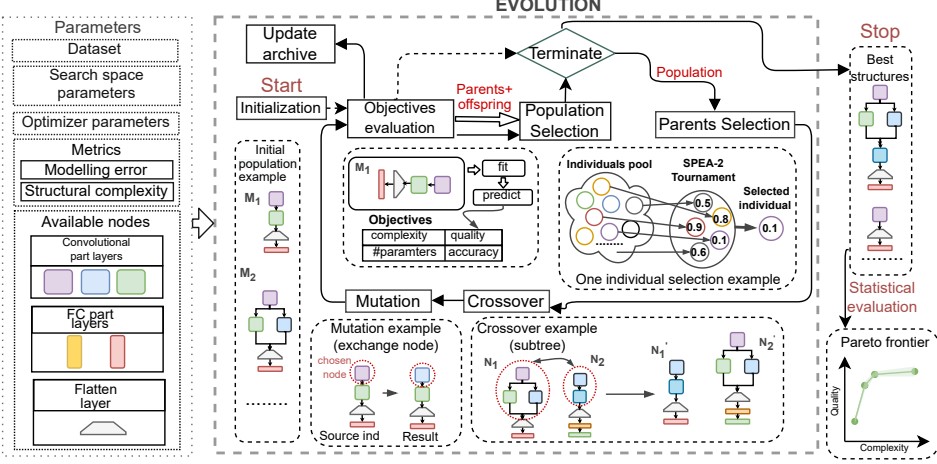

Figure 9: NAS pipeline: evolution, followed by statistical evaluation

## A.8 CONVERGENCY, DIVERSITY AND SPACE COVERAGE TEST

An example of fitness dynamics during evolution is presented in Figure 10.

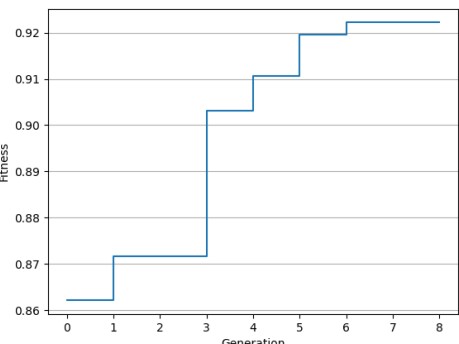

Figure 10: Best fitness dynamics during evolution for CNN+KAN setup on EuroSAT

To ensure coverage of the range of the number of parameters in question chosen according to Appendix B.4, the histograms of the distributions of the numbers of parameters of individuals across evolution generations have been formed.

Note that only the individuals accepted for generations are recorded in the histograms, not all the evaluated candidates.

Together with the Pareto fronts, histograms demonstrate that the complexity range for low- to high-performing models is covered and that architectures with a larger number of parameters have been examined but were unable to generalize well and outperform the smaller models of the resultant Pareto front; hence, exhaustiveness is held.

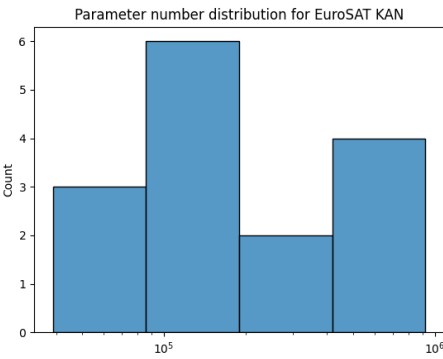

Figure 11: KAN on EuroSAT parameter number distribution histogram

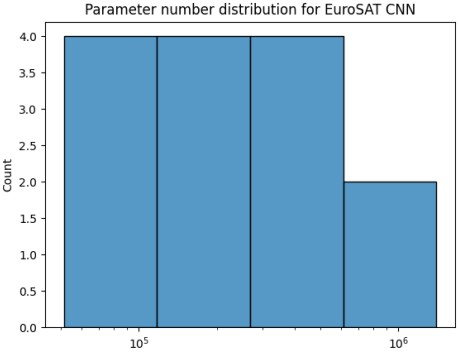

Figure 12: CNN on EuroSAT parameter number distribution histogram

To assess the extent to which the evolutionary process explores various trade-offs between the complexities of the convolutional module and the classification head, we analyze the distribution of parameter shares in the convolutional layers. Figures 14, 15 ,16, and 17 present histograms of the fraction of parameters allocated to the convolutional layers for different architectures. In these histograms, the x-axis demonstrates the fraction of the total number of parameters in the convolutional layers, while the y-axis shows the fraction of the evaluated models exhibiting this property.

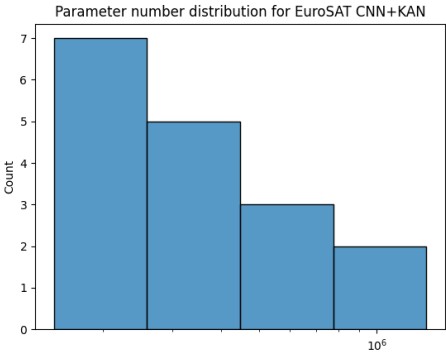

Figure 13: CNN-KAN on EuroSAT parameter number distribution histogram

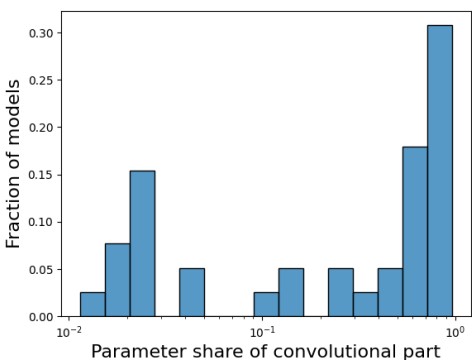

Figure 14: Distribution histogram of parameter shares in convolutional layers for CNN on EuroSAT

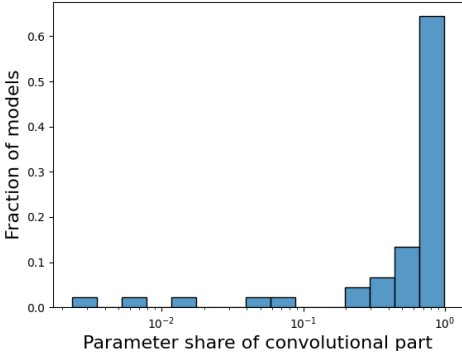

Figure 15: Distribution histogram of parameter shares in convolutional layers for CNN+KAN on EuroSAT

## B APPROACH TO COMPARISON

### B.1 EXHAUSTIVENESS PRINCIPLE

The experiments' guiding principle is to ensure that the evolutionary process is as exhaustive as possible, thereby enabling the comparison of architectural properties rather than random initializations of evolutionary processes.

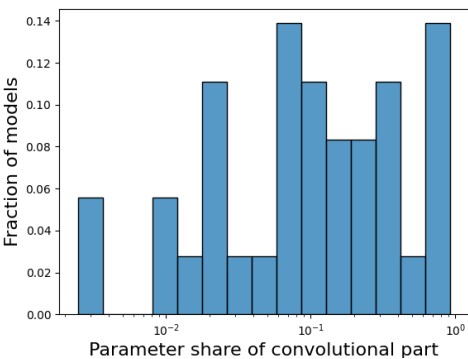

Figure 16: Distribution histogram of parameter shares in convolutional layers for KAN on EuroSAT

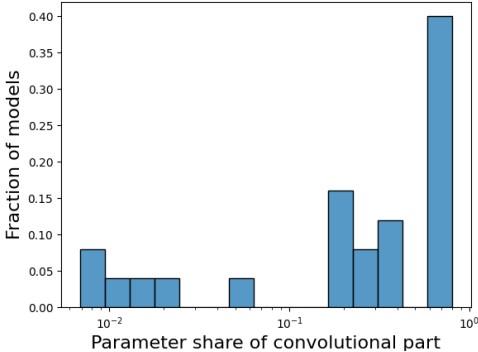

Figure 17: Distribution histogram of parameter shares in convolutional layers for denseConvKAN on EuroSAT

The following measures have been adopted to implement this principle:

- The search space is narrowed by introducing inductive biases specific to each type of structure (CNN, KAN, CNN+KAN).

- Smaller models are discovered to save computational resources, allowing for more trials. Another benefit of smaller models is the more straightforward exploration of a graph space with orders of magnitude smaller than larger models.

- Easier benchmark datasets are selected to suit small models' evaluation better. The principle here is that the tried range of model sizes should correspond to a dataset in a way that covers the full quality improvement interval. This interval typically falls within the set of smaller models for simpler datasets.

While limiting model size reduces computational cost, it may also restrict the discovery of high-performing architectures. We mitigate this by allowing broader kernel size variations and flexible layer depths.

## B.2 ACCOUNTING COMPLEXITY METRIC

As KAN layers are typically drop-in replacements of their NN analogs, a straightforward approach to comparison is to compare the initial NN model with its KAN counterpart with the replaced layers. However, this approach is unfair since KAN layers typically contain more parameters than the replaced ones and require more computation volume and time for both learning and inference.

Popular approaches compare the quality metrics of MLPs and their KAN alternatives with a similar complexity metric, such as the number of parameters or their computational volume, which could be measured in FLOPs.

However, the appropriateness of the comparison technique remains debatable. Judging an architecture solely by its computational volume required for training/inference is not particularly appropriate because models' performance in real-world scenarios is usually constrained more by data availability than by computational resources. A large model can easily learn the training set, but the crucial question is whether it generalizes well, for which fewer parameters are preferred.

Comparing generalization by the number of parameters *across* different architectures is also flawed because of the different ways how parameters affect the transformations. Furthermore, neural networks are known to be over-parameterized to facilitate better optimization landscapes Zhang et al. (2017).

An alternative approach is comparing single best architectures without complexity constraints, as long as they generalize well on the dataset (which is already verified through the evaluation on the test dataset).

Nevertheless, examining the entire Pareto front can be informative, and this metric is helpful for multi-objective optimization. Additionally, a few parameters might indicate better generalization for different feature space distributions close to the training distribution, a property not verified by the validation procedure.

Given the ambiguity in determining the correct approach, we conducted multi-objective optimization based on the number of parameters, and the best models found for the dataset were also compared across different architectures.

### B.3 Params-To-Flops ratio

Despite a typically low parameter number share of convolutional layers in ConvKAN's geometry (sparse geometry), the number of computations of convolutional layers (more than for linear layers) is comparable with their CNN counterparts. The reason is a higher computations-to-parameter ratio.

To illustrate this observation, consider a setup where a convolutional layer has an input dimension of $c$ channels and an output dimension of $cn$ channels, with a kernel size of $k$ and an input/output matrix size of $w \times h$.

$$\frac{\text{computations}}{\text{parameters}}(\text{denseConv}) = \text{const}\,\frac{\overbrace{cn}^{\text{\#output channels}} \cdot \overbrace{ck^2}^{\text{\#elements affecting output pixel}} \cdot \overbrace{wh}^{\text{\# elements in each feature map channel}}}{\underbrace{k^2}_{\text{kernel elements}}\,\underbrace{nc^2}_{\text{\#kernels}}} = \text{const}\cdot wh$$

$$\frac{\text{computations}}{\text{parameters}}(\text{sparseConv}) = \text{const}\,\frac{\overbrace{cn}^{\text{\#output channels}} \cdot \overbrace{k^2}^{\text{\#elements affecting output pixel}} \cdot \overbrace{wh}^{\text{\# elements in each feature map channel}}}{\underbrace{k^2}_{\text{kernel elements}}\,\underbrace{n}_{\text{\#kernels}}} = \text{const}\cdot c\cdot wh$$

The computation-to-parameter ratio is $c$ times higher for sparse geometry. It grows linearly with the number of input channels, which may be high in the last layers of the network, while the ratio remains constant for dense cases.

Such discrepancy suggests that CNNs may accumulate more information per computation volume unit.

This is demonstrated by examining the convolution part of a ConvNet (big), a CNN from ConvKAN, and an analogous sparse KAN network. The convolutional parts consist of two convolution layers with 32 output channels, followed by pooling and two convolution layers with 64 output channels. Tab 18 and Tab 19 list the columns needed to verify this claim for all four convolutional layers of CNN and KAN correspondingly.

| Layer | FLOPs | Parameters | FLOPs/Parameters | FLOPs/Parameters/Pixel |
|-------|-------|------------|------------------|------------------------|
| conv1 | 627.2k | 832 | 753.846 | 0.962 |
| conv2 | 20.1M | 25.6k | 783.021 | 0.999 |
| conv3 | 3.6M | 18.5k | 195.322 | 0.997 |
| conv4 | 7.2M | 36.9k | 195.66 | 0.998 |

Figure 18: Parameter-to-flops ratio for ConvNet (big) from ConvKAN paper

| Layer | FLOPs | Parameters | FLOPs/Parameters | FLOPs/Parameters/Pixel |
|-------|-------|------------|------------------|------------------------|
| conv1 | 5.6M | 8.0k | 705.6 | 0.9 |
| conv2 | 5.6M | 250 | 22579.2 | 28.8 |
| conv3 | 1.0M | 180 | 5644.8 | 28.8 |
| conv4 | 1.0M | 90 | 11289.6 | 57.6 |

Figure 19: Parameter-to-flops ratio for analogous KAN

The FLOPs/Parameters/Pixel ratio remains constant in CNN but raises with the layer depth in KAN.

## B.4 END-TO-END VS TAILORED SEARCH SPACE

Ideally, a unified search space encompassing appropriate ranges for both KAN CNN would be developed as part of an end-to-end approach. However, as evident from the space encoding, the current algorithm cannot automatically narrow down and focus on specific subspaces pertinent to each architecture.

Therefore, an end-to-end approach proves computationally infeasible. Instead, distinct search spaces are designed for CNN and KAN, ensuring that the maximum number of parameters and computational resources are equal. Exact equality is infeasible and unnecessary since the limits are not reached in the resultant Pareto fronts.

The search space is chosen based on the alignment of parameter counts across CNN and KAN models. It covers a sufficiently wide complexity range, as demonstrated by the resultant Pareto fronts: the smallest models achieve low accuracy values, which decline rapidly with size reduction, whereas the larger models exhibit a slow increase in accuracy with size growth.

This indicates that the selected subset of model parameter sizes corresponds appropriately to the dataset complexity.

