# OpenReview forum: "Exploring convolutional KAN architectures with NAS"
_mathai.club/MathAI/2025/Conference — MathAI 2025 Oral_

### Official Review · Reviewer_NW17 · 2025-02-26
**The work solves the important problem of searching and comparing the architectures of Kolmogorov-Arnold networks and convolutional neural networks and can be recommended for publication.**

**Rating:** 9
**Confidence:** 5

**Review:**

Kolmogorov-Arnold networks (KANs), as an alternative to traditional deep learning, offer new perspectives for neural networks, but are understudied and their benefits are difficult to evaluate. This paper presents a Neural Architecture Search (NAS) system that systematically optimizes and compares KANs with convolutional neural networks (CNNs), eliminating human bias in the design. NAS optimizes both architectures under the same constraints, focusing on accuracy, parameter efficiency and computational cost. Therefore, the work is relevant and has theoretical and practical importance.
What is new in the work is the use of Neural Architecture Search (NAS), which provides more objective comparison than using manually designed architectures
The work introduces new KAN architectural components: dense, sparse KAN convolutions and transposed KAN layers, allowing KAN to handle spatial dependencies, and upsampling, previously understudied in the KAN literature.
What is new is the extension of KAN for non-standard applications using the example of sea ice prediction with physical constraints.
This work demonstrates that KANs can compete with CNNs in terms of accuracy but are less efficient, requiring architectural improvements.
The work is characterized by a clear presentation and good structuring.
A drawback is the use of evolutionary algorithms in NAS, which are computationally expensive and limit the search space and model sizes (as noted in Section 4.1).
Overall, the work is of great practical and theoretical importance and can be recommended for publication.

---

### Official Review · Reviewer_YCAq · 2025-02-27
**The work solves the important problem of searching and comparing the architectures of Kolmogorov-Arnold networks and convolutional neural networks and can be recommended for publication.**

**Rating:** 9
**Confidence:** 5

**Review:**

Kolmogorov-Arnold networks (KANs), as an alternative to traditional deep learning, offer new perspectives for neural networks, but are understudied and their benefits are difficult to evaluate. This paper presents a Neural Architecture Search (NAS) system that systematically optimizes and compares KANs with convolutional neural networks (CNNs), eliminating human bias in the design. NAS optimizes both architectures under the same constraints, focusing on accuracy, parameter efficiency and computational cost. Therefore, the work is relevant and has theoretical and practical importance. What is new in the work is the use of Neural Architecture Search (NAS), which provides more objective comparison than using manually designed architectures The work introduces new KAN architectural components: dense, sparse KAN convolutions and transposed KAN layers, allowing KAN to handle spatial dependencies, and upsampling, previously understudied in the KAN literature. What is new is the extension of KAN for non-standard applications using the example of sea ice prediction with physical constraints. This work demonstrates that KANs can compete with CNNs in terms of accuracy but are less efficient, requiring architectural improvements. The work is characterized by a clear presentation and good structuring. A drawback is the use of evolutionary algorithms in NAS, which are computationally expensive and limit the search space and model sizes (as noted in Section 4.1). Overall, the work is of great practical and theoretical importance and can be recommended for publication.

---

### Official Review · Reviewer_qY8P · 2025-02-27
**The article provides a comparison of Kolmogorov-Arnold Networks (KAN) and Convolutional Neural Networks (CNN) using Neural Architecture Search (NAS). The study is relevant for understanding the trade-offs between accuracy and efficiency in neural networks, though it needs more detailed parameter descriptions and clearer appendix references.**

**Rating:** 8
**Confidence:** 4

**Review:**

Review
The article discusses the use of a Neural Architecture Search (NAS) framework to compare Kolmogorov-Arnold Networks (KAN) and Convolutional Neural Networks (CNN) in solving benchmark tasks and one practical task—predicting sea ice concentration. It is shown that KANs can achieve the accuracy and efficiency of CNNs, but only when using a larger number of parameters. As a result, KAN models require significantly more computational resources, making them less efficient compared to CNNs. To reduce the efficiency gap with CNNs, the article considers the use of dense convolutional KANs as a way to address some of the architectural limitations of KAN models. It is noted that automated architecture search (NAS) allows for the identification of non-trivial architectural solutions, reducing the influence of researchers' subjective preferences.
Relevance of the Topic
Kolmogorov-Arnold Networks (KAN) represent one of the alternatives to modern neural network approaches, and some researchers hope that the development of KANs could lead to breakthroughs in the field of machine learning. To assess the validity of such hopes, systematic and unbiased comparisons are required, which is the aim of this study. In the context of limited computational resources, it is important to understand which architectures can provide the best balance between accuracy and computational complexity. The study of KANs and their comparison with CNNs also has theoretical value, as it allows for a better understanding of how different approaches to data modeling affect the performance and generalization ability of models.
Recommendations for Article Revision
For a better understanding of the results obtained in the study, a more detailed description of the number of parameters per layer should be provided, at least for the main compared architectures (CNN, KAN, CNN+KAN). This information can be included in section 4.1 or in Appendix A.
In Appendix B.3, the two mentions of Table B.3 should be differentiated, and captions should be added to the corresponding tables. Additionally, in Appendix B.4 (if the provided interpretation of KAN as Knowledge-Augmented Networks is not an error), an explanation should be given as to why a different interpretation of the widely used abbreviation in the article is used.

---

### Decision · Program_Chairs · 2025-03-08

**Decision:**

Accept (Oral)

**Comment:**

Your article has been accepted and you can give a talk on the article. All articles will be sorted by rating and within the available conference places one author from each article will be invited. If there are not enough places, then you will either have the opportunity to speak remotely or come at your own expense!